# Alkali-Activated Slag Coatings for Fire Protection of OPC Concrete

**DOI:** 10.3390/ma16237477

**Published:** 2023-12-01

**Authors:** Andrius Kielė, Danutė Vaičiukynienė, Šarūnas Bertašius, Pavel Krivenko, Rėda Bistrickaitė, Vytautas Jocius, Dainius Ramukevičius

**Affiliations:** 1Faculty of Civil Engineering and Architecture, Kaunas University of Technology, Studentu St. 48, LT-51367 Kaunas, Lithuania; sarunas.bertasius@ktu.lt (Š.B.); reda.bistrickaite@ktu.lt (R.B.); 2Glukhovsky Scientific Research Institute for Binders and Materials, Kyiv National University of Construction and Architecture, Povitroflotskyi Prospect 31, 03037 Kyiv, Ukraine; pavlo.kryvenko@gmail.com; 3Fire Reseach Center, LT-03223 Vilnius, Lithuania; vytautas.jocius@vpg.lt; 4The Department of Agricultural Engineering and Safety, Vytautas Magnus University, LT-44248 Kaunas, Lithuania; dainius.ramukevicius@vdu.lt

**Keywords:** alkali-activated slag, fire-resistant coatings

## Abstract

During a fire, ordinary Portland cement (OPC) systems lose their mechanical properties. For this reason, it is important to find a way to protect it. This study suggested alternative uses of slag and phosphogypsum to produce coatings for fire-resistant applications. Five compositions of 10 mm thick alkali-activated slag coatings were investigated. In these compositions, different amounts of phosphogypsum (1%, 3%, 5%, 7%, and 10%) were used. In the first stage of this study, the residual compressive strength of samples with the coatings based on alkali-activated slag was compared to the results of OPC concrete samples without coatings. The experimental results showed that a higher residual compressive strength of 33.2–47.3 MPa OPC concrete was achieved for the samples with coatings. Meanwhile, the residual compressive strength of the uncoated samples was 32.37 MPa. In the second stage, OPC concrete samples were reinforced with fiberglass polymer (FRP) rods, and they had a similar positive effect on alkali-activated coatings. After exposure to higher temperatures, the pullout tests of the glass FRP bars showed that the adhesion strength was (9.44 MPa) 43.9% higher for the samples with coatings compared to the samples without coatings (6.56 MPa). Therefore, a higher bond strength can be maintained between concrete and FRP bars. Alkali-activated slag with 3% phosphogypsum can be used for the production of fire-resistant coating. These coatings could protect OPC concrete and reinforced concrete with glass FRP bars from fire.

## 1. Introduction

Ordinary Portland cement (OPC) concrete is a non-combustible material (i.e., it does not burn) and has a slow rate of heat transfer, but this material, like other non-combustible material, can be destroyed by fire. The main purpose of concrete significant strength loss could be calcium oxide, which was formed at elevated temperatures. Khoury et al. [1] stated that concretes based on Portland cement lose their mechanical properties at temperatures above 550–600 °C. OPC concrete becomes structurally useless at temperatures above 800 °C due to the degradation of hydration products [2]. For this reason, the fire protection of reinforced OPC concrete structures is very important worldwide. There are several ways to protect OPC concrete structures. The incorporation of polypropylene fibers led to improved fire resistance of high-strength concretes based on OPC [3]. At 200–250 °C, the fiber helps to release moisture from the concrete, which reduces explosive spalling. Another way to protect OPC concrete structures is to create thermal barriers or coatings that reduce heat flow into OPC concrete. One of the fire protective coatings could be alkali-activated materials. The behavior of alkali-activated binding materials is different compared with ordinary Portland cement binding materials. The fire resistance could be explained by the microstructure of alkali-activated materials. During fire, the moisture is removed from the material via nanopores without any destruction [4]. Cheng et al. [5] stated that for fire resistance tests, a 10 mm thick metakaolinite geopolymer panel was exposed to a flame of 1100 °C, and the measured reverse-side temperatures reached less than 350 °C after 35 min. It has also been reported [6] that when suitable resistance to high temperature is required, geopolymers are considered highly competitive materials thanks to the intrinsic thermal resistance of their structure. For this reason, the fly ash-based geopolymers, activated at room temperature, could be used as fireproofing coatings. Payakaniti et al. [7] determined that the compressive strength of alkali-activated high-calcium fly ash is closely related to the temperatures that were burned. It is important to note that at the temperature of 200 °C, the compressive strength increased due to the geopolymerization reactions. Up to the temperature of 600 °C, the compressive strength of samples remains the same. At a temperature of 800 °C, new phases formed in the alkali-activated fly ash samples. The reduction in mechanical properties could be the degradation of geopolymer gel. The alumosilicate adhesives based on analcime and zeolite ZK-14 phases could be used in fire-resistant structures, as stated by Krivenko et al. [8]. In this case, metakaolin and amorphous silica were activated with soluble sodium silicate, sodium, and potassium hydroxide solution. Quartz sand, chamotte fines, and mica were added as fillers. The stability at high temperatures could be related to the formation of anhydrous phases such as nepheline and leucite. Shih et al. [9] investigated alkali-activated porous furnace slag. These samples had densities in the range of 594 and 1184 kg/m^3^ with compressive strength 0.95 to 9.04 MPa, respectively. The firing tests showed that the plates made from this porous material had better fire resistance than usual rock wool. The back side of the panel had no more than 100 °C temperature; meanwhile, the front panel side had 800 °C temperature (under flames) after 1 h. 

Previous studies on the use of alkali-activated materials for coatings that protect OPC concrete from fire have shown great potential for this type of coating. Protective coatings are used to insulate and protect the surfaces of Portland cement concrete. Krivenko et al. [10] experimentally determined that the optimal thickness of aluminosilicate coating is 6 mm, and it could protect OPC concrete from destruction for at least 2 h. This aluminosilicate coating was made from metakaolin, which was activated with a mixture of sodium silicate and NaOH solution. Rotten-stone pellets and limestone powder as expanding agents and, at the same time, as fillers were incorporated as well. Mohd et al. [11] investigated fly ash geopolymer coating samples. The samples of the geopolymer coating harden to a glassy structure and are effectively used to create a thermal-resistant surface. This technology creates a geopolymer blend design that is suitable for use as OPC systems coatings with high thermal applications. The results [12] obtained have shown that the ability of geopolymers to protect concrete systems from damage at high temperatures has been evaluated by the application of two different geopolymeric coatings to mortar cubes, one based on metakaolin and another based on metakaolin with amorphous silica. Experimental results show that the foaming effect observed in the samples with amorphous silica may be responsible for the 54% retention of compressive strength at 800 °C, while uncoated samples retained only 32%. This difference is significant in cases where the protection of lives is required. Papakonstantinou et al. [13] deal with the performance of a new inorganic fireproof coating based on geopolymers. One of their advantages is that the vapor pressure is released because they do not form a film. When concrete is reinforced, the matrices form a strong bond between the concrete surface and the fiber reinforcement. Temuujin et al. found that [14] geopolymer-type coatings were prepared from the mixture of sodium silicate solution (SiO_2_:Na_2_O = 3.1) and metakaolin. These coatings were applied as a thermal barrier for concrete structures. After calcination at 1000 °C for 1 h, the coatings had an X-ray amorphous structure. Thermogravimetric analysis of two geopolymer compositions showed suitable thermal stability of this material at temperatures of up to 1000 °C. These materials had. Sakkas et al. [15,16,17,18] presented the design of a fire-resistant coating for passive fire protection of a tunnel and its operation under thermal load. The substances belong to the class of potassium-based geopolymers. This K-geopolymer maintained its structural integrity after testing without significant macroscopic damage [15]. In another research [16], the geopolymer was prepared by mixing metakaolin, with the addition of solid SiO_2_, with a highly alkaline aqueous phase of potassium hydroxide to form a paste, which was then cured for some time at 70 °C. The incorporation of solid SiO_2_. The addition of solid SiO_2_ was important to make the geopolymer fire resistant at temperatures up to 1350 °C. After the test, the surface of the geopolymer was scrubbed and cracked without losing structural integrity. Thus, the concrete slab protected by the geopolymer did not experience any form of compression or a decrease in its compression strength. In the later research [17], the K-geopolymer-based material was prepared by mixing ferronickel (FeNi) slag with pure alumina filled with a highly alkaline KOH aqueous solution. During and after the two tests, the material maintained the integrity of the structure without visible any mechanical or thermal damage. During the passive fire protection test, it was concluded that investigated geopolymer materials had excellent fire-resistant properties and could be used for the protection of OPC concrete constructions [18]. Privorotskaya [19] determined that the coating based on metakaolin-based geopolymer with the addition of amorphous silica had a positive influence on the strength values at higher temperatures. At 450 °C, the mortar reached 12% and 14% lower reduction in strength for the samples with this type of coating compared with samples without coating. At the temperature of 800 °C, the reduction in strength was more than 20%. It has been found that metakaolin-based geopolymer could be used as a valuable coating for fire protection of OPC concrete. In another study [20], the data related to the fire resistance of alkali-activated coating materials were summarized, and a conclusion was drawn that these coatings have a high fire-retardant potential for OPC systems. Increasing the fire resistance of these coatings is closely related to the formation of special minerals in alkaline materials, and it depends on the use of special additives.

The main properties of alkali-activated materials or geopolymers from which fire protection coatings are made are closely related to geopolymerization and hydration mechanisms. After the aluminosilicate precursor was filled with alkali solution, the activation process started. During the geopolymerization reactions, precursors of the active form were dissolved in the alkaline solution, and new aluminosilicate phases were formed. Zeolitic crystalline compounds, C–S–H and C–A–S–H, and amorphous alumino-silicate structures are detected in the slag hydration products [21,22]. These new hydration zeolitic compounds are durable at elevated temperatures [23]. Shilar et al. [24] concluded that the hydration rate of ground granulated blast furnace slag decreases when the molarity of the alkali activator increases from 8 M to 12 M. The optimal hydration temperature was found to be 70 °C. In another study [25], the best ratio of sodium silicate solution to precursor made from metakaolin and fly ash was 0.7. In this case, the highest compressive strength reached 40.2 MPa.

In this study, the original composition of alkali-activated slag was suggested. The aim of this research was to investigate the potential use of alkali-activated slag as a coating to protect concrete samples without reinforcement and with glass fiber polymer reinforcement in high-temperature applications. An alkali–alkali-activated slag system with phosphogypsum was chosen because both initial materials are by-products. Large amounts of the phosphogypsum are accumulated, and only a small amount of them is reused. The aim is to reduce the amount of waste in the context of the circular economy.

## 2. Experimental

### 2.1. Raw Materials

In this study, Portland cement CEM I 42.5 R was used as a binding material for the production of concrete samples. This cement was produced by the company “Akmenes cementas” (Naujoji Akmenė, Lithuania). The initial setting time of Portland cement was 140 min, and the final setting time was 190 min [26] (EN 196-3). The mineral composition of Portland cement contained C3S–62.0%, C2S–12.0%, C3A–7.5%, and C4AF–11.0% (according to X-ray fluorescence analysis). The specific surface of this cement was 229 m^2^/kg. The 0/4 fraction sand from the Kvesai quarry was used as fine aggregate, and 5/16 fraction granite macadam was used as coarse aggregate. Concrete samples—cubes (100 × 100 × 100) mm were formed using the compositions from Table 1. These samples were cured in conditions according to standard EN 12390-2:2019 [27] and tested after 28 days of hydration. The compressive strength of concrete samples was determined according to standard EN 12390-3:2019 [28]. The ratio of water–cement was 0.43. Other constituents are provided in Table 1.

In order to explore the effectiveness of alkali-activated slag coatings, ordinary Portland cement (OPC) concrete samples were covered with 50 mm thick alkali-activated materials. Three samples were formed for each mixture. The properties of the (OPC) concrete samples with alkali-activated slag coatings were compared with the properties of reference samples without alkali-activated slag coatings. All samples were cured under the same conditions.

Alkali-activated slag coatings are produced using slag, phosphogypsum, and sodium hydroxide solution as alkali activators. In the milled slag (207 m^2^/kg), the oxides of CaO and SiO_2_ predominated: CaO is 45.20%, SiO_2_ is 37.10%, Al_2_O_3_ is 6.44%, MgO is 5.76%, SO_3_ is 1.85%, and other chemical element oxides in amounts not exceeding 1% (according to X-ray fluorescence analysis). Semi-hydrate phosphogypsum generated by fertilizer plants was used as well. The main mineral of phosphogypsum was CaSO_4_ 0.5H_2_O (up to 98%), and in this case, chemical analysis methods were used. However, there are impurities of phosphorus and fluoride compounds. The five types of alkali-activated slag coatings were prepared in this research (Table 2). The amount of phosphogypsum (0%, 1%, 3%, 5%, 7%, and 10%) varied in each type of coating. First, the dry initial materials, such as slag and phosphogypsum powder, were carefully mixed. Later, the dry mixtures are poured with sodium hydroxide solution. The mass was mixed again and finally covered OPC concrete samples.

After one day of hydration, the reference Portland cement concrete samples were removed from the molds (demolded) and covered with polyethylene material, which protects the water evaporation. Other samples were covered with alkali-activated slag coatings, as shown in Figure 1, and then covered with polyethylene covering material as well.

On the first curing day (24 h), samples coated or uncoated with alkali-activated slag coatings were cured at ambient temperature (22 °C). On the second day, after 24 h, samples were cured at the temperature of 60 °C in the oven, and finally, they were cured again at ambient temperature for the last 26 days. After 28 days of hydration, samples were exposed to the temperatures caused by fire for up to three hours to evaluate the potential of alkali-activated coatings for fire protection of Portland cement concrete.

Some of the OPC concrete samples were reinforced with fiberglass polymer (FRP) rods. The diameter of the reinforcement used for the tests was 10 mm.

### 2.2. Test Methods

X-ray fluorescence analysis (XRFA) and chemical composition analysis were used for the determination of the oxygen composition of initial materials such as OPC, slag, and phosphogypsum. The XRFA analysis of slag was performed using a fluorescence spectrometer S8 Tiger (Bruker AXS, Karlsruhe, Germany) operating at the counter gas helium 2 bar. This spectrometer is equipped with a Rh tube with an energy of up to 60 keV. Powder samples (passed through a 63 μm sieve and pressed to cylindrical tablets of 5 mm × 40 mm) were measured in He atmosphere, and the data were analyzed with SPECTRAPlus QUANT EXPRESS standardless software [29].

The chemical composition of phosphogypsum was determined by the classical methods of chemical analysis according to the standards (EN 196-2 [30]; GOST 20851.2-75 [31]). The specific surface area was measured using the Blaine (air permeability) apparatus according to the EN 196-6:2018 standard [32]. Among the many tests applied to concrete, the most important is the compressive strength test, which gives an idea of the concrete’s properties. The compressive strength of samples was tested using a hydraulic press Toni Technik 2020 after a fire resistance test. The compressive strength of concrete samples after fire testing was performed according to standard EN 12390-3:2019 [28]. Compressive strength was calculated from three identical cubes (100 mm × 100 mm) as an average value. After the fire test, the same standard that we used for the compressive strength of the concrete samples before the fire test was used.

The one-sided test method was used for the fire resistance of concrete samples. After 28 days of hydration, fire tests were performed on the concrete samples. The holes for thermocouples were drilled in the concrete samples with a protective layer. Three holes 3 mm in diameter and 50 mm deep were drilled in each sample. K-type thermocouples were used to determine temperatures in different concrete layers of the heated plane. The recording time of the thermocouple temperature change was 3 s. Thermocouples were installed in the center of the sample to measure the temperature. The thermocouple holes were spaced 15 mm apart. The scheme of the sample with thermocouples is shown in Figure 2.

The fire resistance test was carried out according to the standard prEN 1364-1 rev [33] in Lithuania for fire resistance testing of incontinence elements. The tests were performed using the oven (Figure 3a), which is covered from one side by a 3000 mm × 3000 mm test wall. The test oven uses gaseous fuel. The thermocouples were installed for the temperature measuring the inside of the oven, according to Equation (1):
(1)$$T=345{log}_{10}(8t+1)+20$$
where *T*—temperature inside the oven, °C; *t*—time, min.

The 3000 mm × 3000 mm wall of the test oven was constructed from expanded clay blocks. The investigated concrete samples were covered with kaolin wool and incorporated into this testing wall (Figure 3b). Finally, the open furnace plane was covered with the test wall with concrete samples. Samples with drilled holes were fitted with thermocouples with cable ends to connect them to a contact. When the furnace was covered, the test wires from the thermocouples were connected to the contact.

In the second part of this study, the OPC concrete samples with reinforcement were investigated. Glass fiber-reinforced polymer (GFRP) rods (Figure 4a) with vinyl ester resin were used (Figure 4a). The shape of this reinforcement was ribbed.

The anchoring length of the reinforcement bars was the same for all samples, and the reinforcement bars were incorporated in the middle of the concrete cube. The ends of the reinforcement were embedded in plastic tubes to form an anchoring portion of a rod of a certain length (*l* = 5 d). Without a mechanical anchor, the heads of the reinforcement bars were investigated [34]. In accordance with the guidelines of the standard ISO 10406-1:2008 [35] for the pullout tests of composite reinforcement, the anchoring length was calculated according to Equation (2):
(2)$$l_{a}=5\emptyset$$
where *l*_a_—the anchoring length of the reinforcing bar, mm; ∅—diameter of the reinforcing bar, mm, was chosen.

Schemes of reinforcement ready for concreting (Figure 3a) and their position in the samples are shown in Figure 3b.

In these experimental studies of adhesion stresses, short anchoring samples were used. In this case, it can be assumed that the adhesion stresses are evenly distributed over the entire length of the anchorage. The schema of pullout tests for GFRP bars is shown in Figure 4b.

During the pullout tests of GFRP bars, the average adhesion stresses acting on the contact zone of concrete reinforcement are calculated according to Equation (3) as follows:
(3)$$\tau_{\mathrm{av}}=\frac{P}{\pi \cdot \emptyset \cdot l}$$
where P —pulling force, kN; ∅—diameter of the reinforcing bar, mm; l—the length of rod adhering part for concrete, mm.

The analysis and test measurements used in the study contribute to the potential use of alkaline-activated slag as a coating to protect unreinforced and reinforced glass fiber polymer concrete samples at high temperatures (Figure 5).

## 3. Results and Discussion

### 3.1. Fire Resistance Tests of Concrete Samples with Alkali-Activated Slag Coatings

Concrete temperature testing at fire-induced temperatures can be used to monitor concrete temperature variations, which are closely related to concrete strength development. Concrete samples (cubes) without and with alkali-activated slag coatings were exposed to the fire-induced temperature test wall for up to three hours (Figure 6). The development of temperature in the oven is shown in Figure 6a. Over three hours, the temperature gradually increased to about 1100 °C. For three hours, the temperature for each thermocouple was measured, and the results are shown in Figure 6b–d.

It was seen that the first thermocouples were mounted closest to the temperature-exposed surface (15 mm), so the highest temperatures (453–783 °C) were recorded compared to the temperatures reached by the second and third thermocouples (Figure 6b). During the fire test, the temperature of the concrete samples without any coating was 783 ℃. This temperature was reached after 70 min. The thermocouple burned due to heat and began to show inaccurate results above this temperature (783 °C). Alkali-activated coatings on concrete samples had a significant effect on the temperature rise inside the samples. The first thermocouples, which were closest (15 mm) to the temperature-affected surface, measured the temperatures of samples 3S and 10S from 453 to 623 °C, respectively. It consists of 20 to 42% of the temperature reached by the sample compared to the temperature of the reference sample after 70 min. The best results were obtained using an alkali-activated 3S coating during all study periods. Comparing the 3S and 10S samples with the reference sample, it was found that the thermocouples installed in the samples with coating materials did not burn after 70 min of testing, while the thermocouples in the reference sample were damaged.

Differences in measured temperatures were lower for other thermocouples, which had a greater distance from the affected surface (Figure 6c,d). The temperature range of the second thermocouples was from 831 °C for the reference samples, and the lowest temperature was 692 °C with the alkali-activated 3S coating. The situation was similar with third thermocouples. In this case, the highest temperature was 347 °C (reference samples), and the lowest temperature was 246 °C for the samples with 3S coating after 70 min. Similar results were obtained after all study periods. The lowest temperature was reached for samples with 3S coatings.

For the comparison of temperatures in different thermocouples after one hour, the values of the measured temperatures are provided in Figure 7. Based on these experimental results, it can be stated that the best coating material is 3S, and in this case, the lowest temperature was achieved in all the investigated thermocouples. A similar study by Rivera et al. [36] investigated heat transfer through the depth of samples based on alkali-activated binders. It was concluded that alkali-activated fly ash had the largest temperature differences at 565 °C compared to the temperature at the surface and the middle of the sample. The low heat transfer could be related to the mineral composition of alkali-activated materials.

### 3.2. The Residual Compressive Strength of Concrete Samples after Fire Resistance Test

The residual strength of concrete is needed to assess the safety of concrete structures after a fire. The potency of concrete samples to withstand elevated temperatures was based on the values of residual compressive strength that were observed before and after the fire resistance test. The evaluation of concrete samples’ compressive strength after the fire resistance test was carried out in order to determine the effect of high temperature on the compressive strength of concrete. Before the fire resistance test (the initial compressive strength), the compressive strength of the reference samples was 56.5 MPa. After fire testing, the residual compressive strength of concrete samples was measured (Figure 8). In all investigated cases, samples with alkali-activated coatings had higher compressive strength compared with the reference samples without coating. The values of residual compressive strength showed that these alkali-activated materials are effective coatings that could be used to protect concrete from fire.

At elevated temperatures, the mineral composition of alkali-activated slag coatings changed. In our previous study [37], mineral composition was analyzed before and after exposure to elevated temperature using X-ray powder diffraction (XRD) analysis. Before the testing of alkali-activated slag coatings, the main phases of calcium silicate hydrate, calcium aluminum silicate hydrate, portlandite, hydrotalcite, calcite, quartz, and sodium sulfate dominated the mineral composition. After exposure to elevated temperatures (1000 °C), akermanite, gehlenite, merwinite, sodium calcium silicate, and sodium sulfate formed in these coatings with inherited high thermal stability.

After three hours of the fire resistance test, the residual compressive strength of the reference sample decreased to 32.37 MPa, which consisted of 57.26% compared to the compressive strength before the fire resistance test. The residual compressive strength of concrete samples was reached by 58.8–83.7% (33.24–47.31 MPa) for samples with alkali-activated slag coatings after the fire resistance test. The residual compressive strength of the concrete samples with 3S protective coating was only 83.69% after the fire resistance test. Similar values of the residual strength (87%) of alkali-activated aluminosilicate binder-based adhesives were determined by Krivenko et al. [38].

Therefore, these concrete samples with alkali-activated slag coatings exhibited better thermal stability than only concrete samples without these coatings. The high-temperature behavior of alkali-activated slag differs from conventional Portland cement systems. The differences are closely related to the mineral composition of these systems. In the composition of an alkali-activated slag system, calcium hydrosilicates modified with Na^+^ ions and alkali–alkali earth hydro aluminosilicates (hydronepheline, analcime, natrolite, and sodalite) formed during the hydration process. These minerals predetermine the smooth course of their dehydration and topotactic recrystallization into anhydrous compounds, such as calcium, magnesium, and plagioclase silicates, without disturbing the structure of the hardened cement stone. In the OPC system, these compounds were not detected. Rovnaník et al. [39] determined that alkali-activated slag has excellent mechanical development at temperatures above 800 °C. At the temperature of 600 °C, akermanite with a small amount of diopside and wollastonite phases were crystallized in the pastes of alkali-activated slag. This change in mineral composition and microstructure could be related to an increase in compressive strength.

The compressive strength of the coatings before and after the fire resistance test was determined in our previous study [37]. This strength depends on the amount of phosphogypsum in the alkali-activated slag system. The highest compressive strength had samples with 3–5% of phosphogypsum.

### 3.3. The Concrete Samples Reinforced with Glass Fiber Polymer Bars after Exposure to Elevated Temperatures

In the production of reinforced concrete, the use of reinforcement made from glass fiber polymer (FRP) bars instead of steel has become more popular [34]. High temperatures caused by fires or climate changes could affect the concrete structures with glass FRP bars. Elevated temperatures have a negative effect on the performance of glass fiber polymer (FRP) bars. These bars do not burn when embedded in concrete. The lack of oxygen will cause the epoxy to soften at elevated temperatures. Together with softening, degradation of reinforcement occurs, and the mechanical properties decrease [40]. For that reason, it is important to know how this reinforced concrete will perform in an environment with elevated temperatures.

In elevated temperatures, the adhesion of glass FRP bars and concrete could change. Initially, the pullout tests of the fiberglass GFRP bars reinforced concrete was determined. This bond strength of the reference sample before exposure to elevated temperatures consisted of 13.3 MPa (Figure 9).

The bond strength behavior of glass FRP bars and concrete decreased at elevated temperatures. After exposure to elevated temperatures, the bond strength of concrete samples without protective coatings (TC) was 56% lower (6.6 MPa) compared to the reference samples before exposure to elevated temperatures (13.3 MPa). The alkali-activated slag coatings had a positive effect on the bond strength of glass FRP bars and concrete. The highest values of bond strength were maintained by the samples with a protective T3% (with 3 wt.% of phosphogypsum, Table 2) coating of concrete. The bond strength of this sample was 29% lower (9.4 MPa) than that of the reference samples (13.3 MPa). Therefore, alkali-activated slag coatings improve the bond strength and were in the range of 7.4–9.4 MPa.

These studies have been confirmed by another research. Glass FRP bars lose their mechanical properties in environments with elevated temperatures. Alsayed et al. [40] determined that the tensile strength of 40 mm thick reinforcement decreased by about 42% after 3 h at a temperature of 300 °C. The bars with concrete cover showed higher residual tensile strength compared to samples without coating. The tensile strength of glass fiber reinforced polymer (GFRP) bars coated with ordinary Portland cement decreased by 35%. Unfortunately, the concrete covering was more effective at low temperatures (such as 100 °C) and at the shortest duration period (1 h). El-Gamal et al. [41] determined that at 350 °C temperature, the bond strength of glass FRP bars and concrete decreased by increasing the exposure temperature or duration. The maximal reductions of about 50% in the bond strength were determined after exposure to 350 °C temperature for durations of 2 and 3 h. Hamad et al. [42] stated that after 3 h at 325 °C temperature, the bond strength between concrete and FRP bars decreased by up to 81.5%.

After pullout tests, all concrete cubes were cut in the middle to check the mode of failure and to compare the color of the GFRP bars (Figure 10). The samples of T3% did not show a black color in the glass GFRP bar. Meanwhile, the largest area of black color was detected at the sample TC without alkali-activated slag coating.

## 4. Conclusions

The test results showed that alkali-activated slag coatings of OPC concrete created effective thermal barriers that could be used for protection from high-temperature environments. At the temperature caused by fire, the development of concrete strength is related to the reached temperatures in concrete. The best results were obtained using an alkali-activated coating with 3% phosphogypsum in all studies. In this case, the lowest temperatures in the OPC samples were measured, and the highest residual compressive strength was determined. OPC concrete samples with the alkali-activated slag coating exhibited residual strength of 33.2–47.3 MPa; meanwhile, the reference sample has 32.4 MPa. The addition of phosphogypsum has a significant effect on the values of residual strength. The optimal amount of phosphogypsum in the alkali-activated slag coatings was 3%. In this case, the highest residual compressive strength was 47.3 MPa, and it consisted of about 84% compared to the compressive strength before the fire resistance test. So, reference samples without coatings had only 57% compressive strength after the fire resistance test.

Alkali-activated coatings had a similar effect on concrete samples reinforced with glass fiber polymer bars that were exposed to elevated temperatures. The highest values of bond strength, between glass FRP bars and concrete, had samples coated with alkali-activated slag with 3% phosphogypsum. This bond strength (9.44 MPa) was higher by 43.9% than the bond strength of the reference sample (6.56 MPa).

It could be stated that alkali-activated slag coatings with the addition of phosphogypsum for 3% could be used as a fire-resistant coating. It increased the fire resistance of Portland cement concrete and improved the bond strength of concrete and glass fiber polymer bars at elevated temperatures.

## Figures and Tables

**Figure 1 materials-16-07477-f001:**
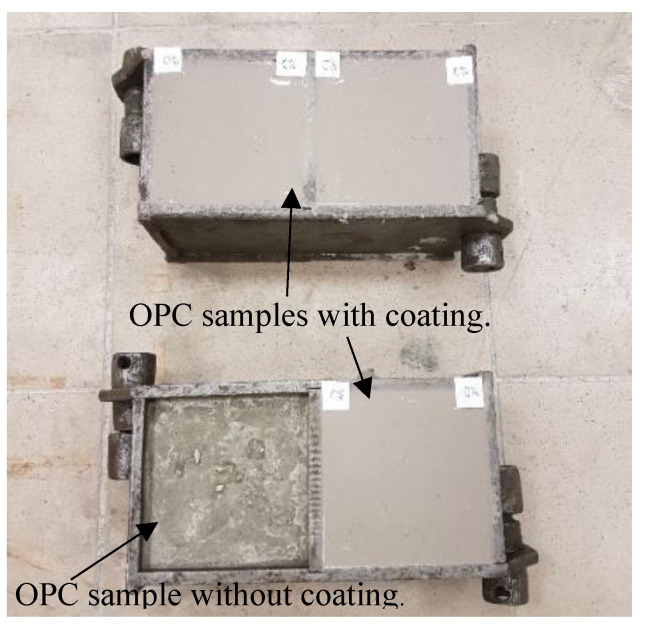
The Portland cement concrete samples covered or uncovered alkali-activated slag coatings.

**Figure 2 materials-16-07477-f002:**
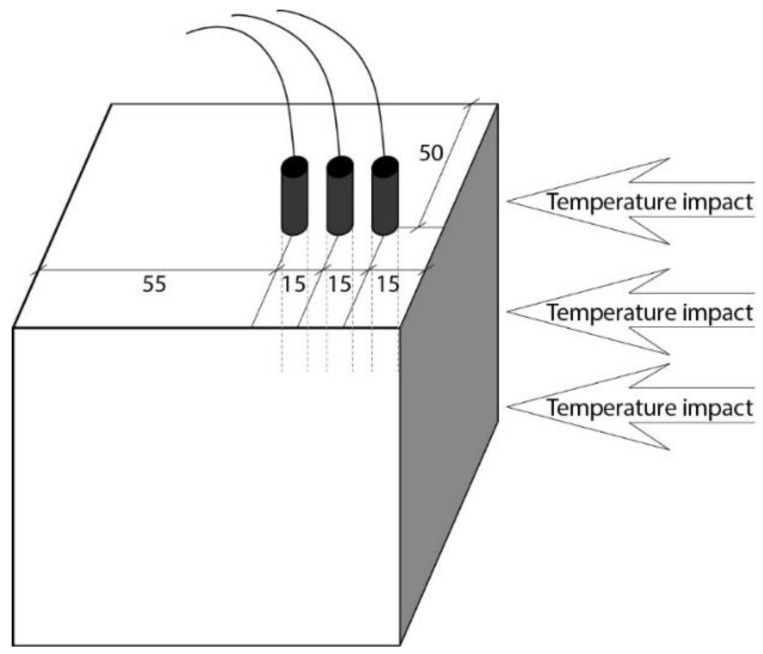
The locations of K-type thermocouples in the center of the concrete sample.

**Figure 3 materials-16-07477-f003:**
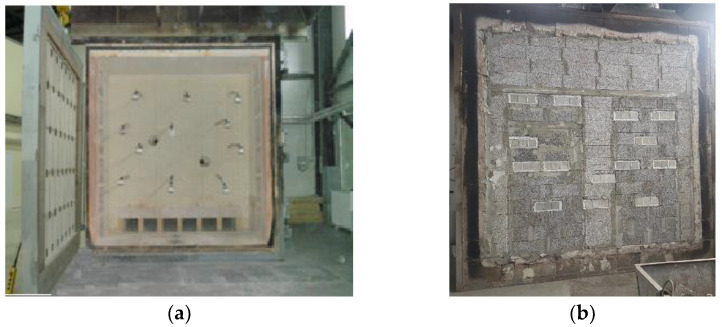
The oven for fire resistance test. Open test wall (**a**) and close test wall with constructed concrete samples in it (**b**).

**Figure 4 materials-16-07477-f004:**
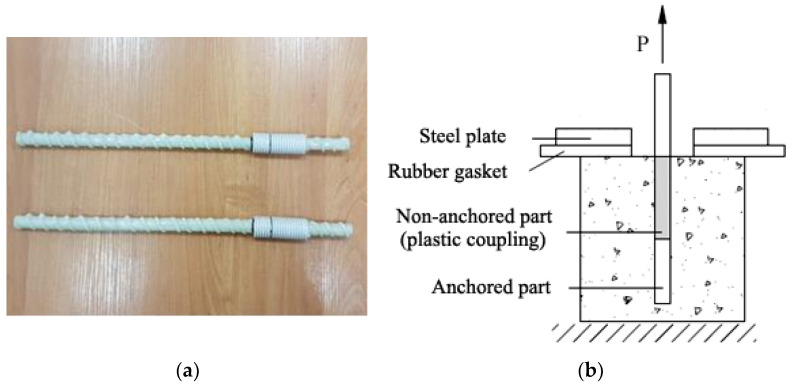
Glass fiber-reinforced polymer bars: the preparation of reinforcement; (**a**,**b**) the schema of the pullout tests of GFRP bars.

**Figure 5 materials-16-07477-f005:**
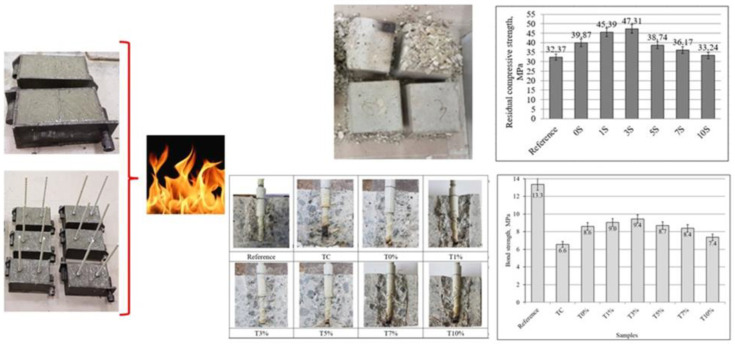
Schematic representation of the study of alkali-activated slag coatings for fire protection of OPC concrete samples.

**Figure 6 materials-16-07477-f006:**
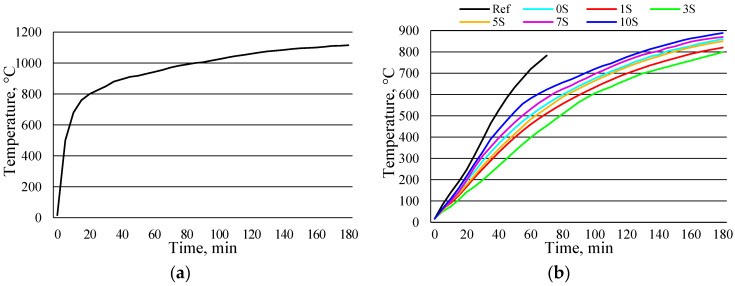
The time–temperature relationships of OPC concrete samples with alkali-activated slag coatings. Notes: the temperature in the oven (**a**); the 1st thermocouple located 15 mm from the temperature-affected surface (**b**); the 2nd thermocouple located 30 mm from the temperature-affected surface (**c**); the 3rd thermocouple located 45 mm from the temperature-affected surface (**d**).

**Figure 7 materials-16-07477-f007:**
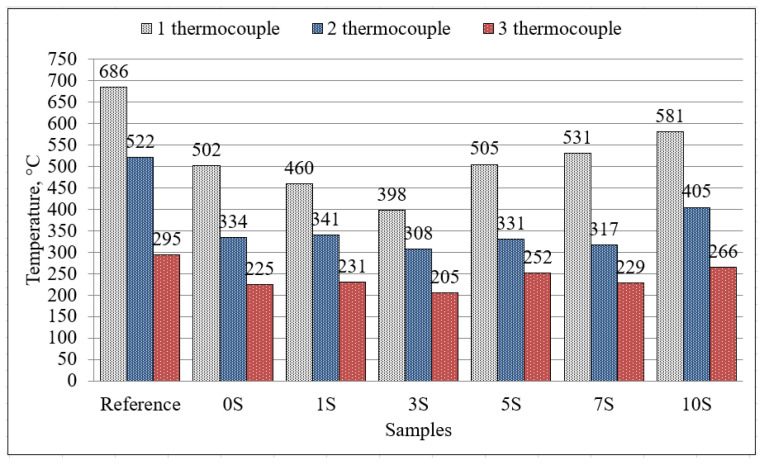
The temperature of samples measured by thermocouples after one hour.

**Figure 8 materials-16-07477-f008:**
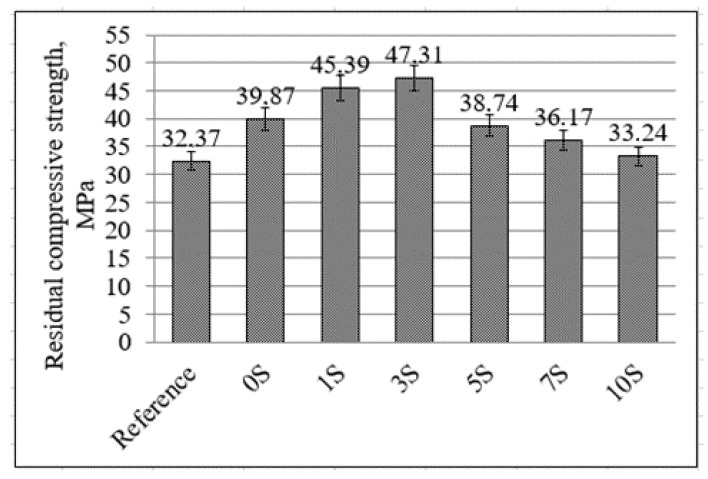
The residual compressive strength of concrete samples with alkali-activated slag coatings after the fire resistance test.

**Figure 9 materials-16-07477-f009:**
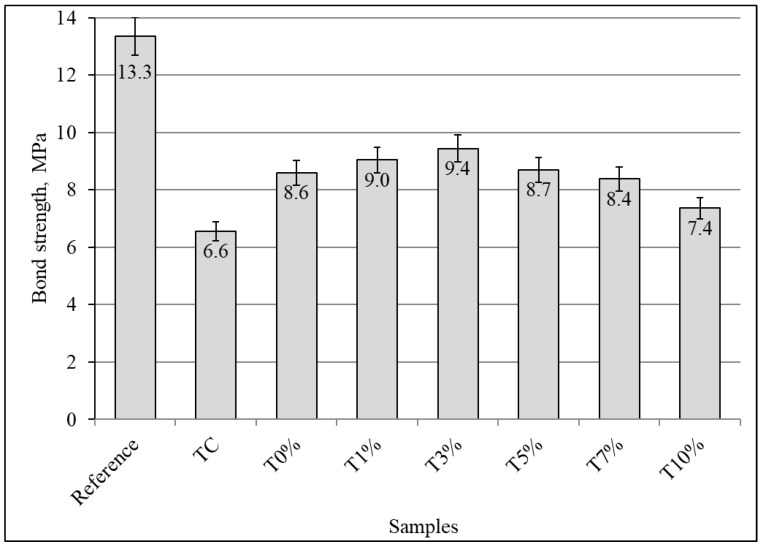
The effect of alkali-activated slag coatings on the pullout tests of the glass FRP bars after the fire resistance test.

**Figure 10 materials-16-07477-f010:**
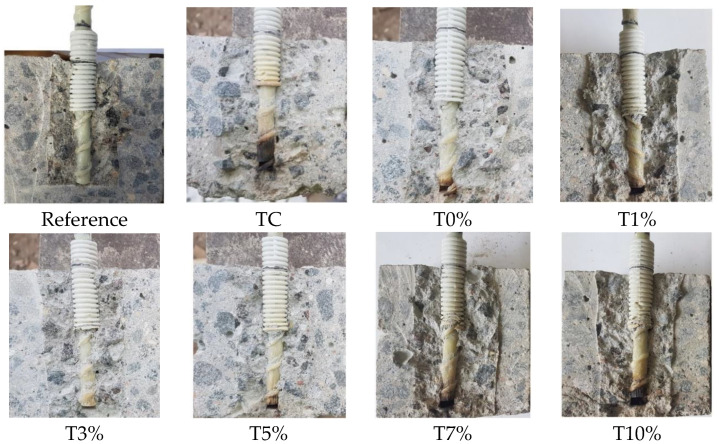
The photos of concrete samples reinforced with glass GFRP bars after exposure to elevated temperatures and after the pullout tests of reinforcement.

**Table 1 materials-16-07477-t001:** The composition of 1 m^3^ of OPC concrete mixtures (concrete mix design was calculated by the absolute volume method).

Materials	Quantities for 1 m^3^ of Concrete Mixtures
Portland cement (C), kg	305.02
Water (W), L	130.25
Fine aggregate (0/4 fraction sand), kg	839.48
Coarse aggregate (5/16 fraction granite macadam), kg	1068.07
Water and cement ratio, W/C	0.43
The density of fresh concrete mix, kg/m^3^	2342.82

**Table 2 materials-16-07477-t002:** The initial mixtures of coatings and their compressive strength.

Coatings	Slag, wt.%	Phosphogypsum, wt.%	Sodium Hydroxide, wt.%	W/S *	Compressive Strength, MPa
0S	100	0	9.71	0.25	38.68
1S	99	1	9.71	0.25	39.53
3S	97	3	9.71	0.25	39.71
5S	95	5	9.71	0.25	49.63
7S	93	7	9.71	0.25	31.12
10S	90	10	9.71	0.25	28.45

* W/S—water solid ratio.

## Data Availability

Data are contained within the article.

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
