# Peer review of "Alkali-Activated Slag Coatings for Fire Protection of OPC Concrete"

_materials, 2023, doi:10.3390/ma16237477_

Round 1

Reviewer 1 Report

Comments and Suggestions for Authors

   This paper studies the fire protection of ordinary Portland concrete by studying alkali-activated slag coatings. The results of the test can also prove that the fire protection achieved by different phosphogypsum levels of alkali active slag coating is also different, and the high temperature tensile test can also reflect that alkali active slag coating can improve the bonding strength of concrete and glass fiber polymer rod at high temperature. It provides a research result worthy of reference for concrete fire protection.

   Regarding the content and conclusion of the article, there are several issues that need to be revised:

1、The test innovation is insufficient, and the test has few variables in terms of simply adjusting the amount of phosphogypsum, and the test scheme should continue to be improved.

2、The background is not clear enough in the introduction, and the literature is overly cited and cluttered. The shortcomings of previous studies should be systematically summarized and combined with the innovations of this trial.

3、The microscopic reasons why alkali-activated residue can effectively prevent fire are not explained enough, and the experimental results are not fully explained.

4、Can the fire effect of alkaline active slag coating and the improvement of bonding strength of glass fiber polymer rod be explained by reasonable microscopic analysis?

5、The result chart of the experiment is not clearly expressed, as shown in Figure 5, you can replace it with a dotted line or dot, otherwise black and white printing will cause the chart results to be unrecognizable. And the vertical axis scale should be consistent, so that readers can more clearly understand the comparison with the experimental results of the reference group. The test result chart is mostly a bar chart, and a line can be added on the basis of the bar chart, which can better reflect the change under different dosages.

6、Some experimental pictures are not clearly expressed, the annotation of Figure 1 is not clear, readers cannot distinguish and compare, it is recommended to make an experimental flow chart to help readers understand.

Overall, the paper verifies that alkaline active slag coatings with 3% phosphogypsum can be used as fireproof coatings. The fire resistance of ordinary Portland cement concrete is improved, and the bonding strength of concrete and glass fiber polymer is improved. The experimental effect is good, but the overall structure and graphic expression of the article need to be greatly modified.

Comments on the Quality of English Language

Moderate editing of English language required

Author Response

Hello,

Reviewer 2 Report

Comments and Suggestions for Authors

Comments on the Quality of English Language

There are a few errors in the English sentences of the thesis, which need to be further revised.

Author Response

Hello,

Reviewer 3 Report

Comments and Suggestions for Authors

In work ALKALI ACTIVATED SLAG COATINGS FOR FIRE PROTECTION OF OPC CONCRETE was discused for alternative uses of slag and phosphogypsum to produce coatings for fire-resistant applications.

The work is acceptable in its current form with minor technical corrections, like:

1. No affiliation after authors,

2. after table 1 and the text put one line space...

Comments on the Quality of English Language

Dear editors,

I agree to accept the paper for publication with minor technical corrections. The paper is clearly written, the results are adequately presented, and the conclusion follows the entire paper.

Author Response

Hello,

Reviewer 4 Report

Comments and Suggestions for Authors

materials-2669596

Title: Alkali Activated Slag Coatings for Fire Protection of OPC Concrete

After careful evaluation, I have concluded that the above-mentioned manuscript requires major revision

Comments

1.      The article's title selection process is intriguing. It's worth exploring how the authors arrived at the title and what led them to employ the term "alkali-activated slag." Furthermore, it's essential to understand the rationale behind the authors' choice and how they justified it.

2.      What instrument was used for XRFA analysis of the slag? Provide more details about the operating conditions of the fluorescence spectrometer S8 Tiger 185?

3.      How was the chemical composition of phosphogypsum determined, and which standards were followed for this analysis? What specific method was used to measure the specific surface area of the samples?

4.      Elaborate on the process and standards followed for testing the compressive strength of the samples? Were there any specific parameters or conditions for the fire resistance test mentioned? What was the purpose of conducting compressive strength tests after fire testing?

5.      Were there any specific standards or guidelines followed for the compressive strength testing of concrete samples after fire testing?

6.      Were there any significant findings or results from the XRFA analysis, chemical composition analysis, specific surface area measurement, or compressive strength testing?

7.      How did the data from these analyses and tests contribute to the overall study or research project? What type of reinforcement bars were used in the concrete samples?

8.      What material was used for the glass fiber-reinforced polymer (GFRP) bars? Were the anchoring lengths of the reinforcement bars uniform? What was the diameter of the reinforcement bars used in the tests?

9.      How were the ends of the reinforcement bars prepared for anchoring?

10.  What standard was followed for the pullout tests of composite reinforcement, and what equation was used for calculating anchoring length?

11.  How were adhesion stresses assumed to be distributed in the anchorage in these experimental studies?

12.  How were the average adhesion stresses calculated during the pullout tests of GFRP bars?

13.  What was the objective of exposing concrete samples to the fire-induced temperature test?

14.  How long were the concrete samples exposed to the fire-induced temperature test? Provide details about the temperature development during the test, as shown in Figure 5a?

15.  How were temperature measurements taken during the test?

16.  What was the reason for placing the first thermocouples closest to the temperature-exposed surface?

17.  What was the highest temperature recorded by the first thermocouples, and how did it compare to the second and third thermocouples? At what point during the fire test did the concrete samples without any coating reach 783 ℃? Provide temperature data for the first thermocouples for both samples 3S and 10S?

18.  How did the temperature measurements for samples 3S and 10S compare to the reference sample after 70 minutes of testing?

19.  Which coating yielded the best results during all study periods, and what were those results? What were the temperature differences among the other thermocouples with greater distances from the affected surface (Fig. 5c, d)? Were the lower temperatures reached by samples with the 3S coating consistent throughout all study periods?

20.  What was the purpose of comparing temperatures in different thermocouples after one hour?

21.  Which coating material yielded the lowest temperatures in all investigated thermocouples?

22.  How was the potency of concrete samples to withstand elevated temperatures determined?

23.  What was the initial compressive strength of the reference samples before the fire resistance test?

24.  Were the concrete samples with alkali-activated coatings found to have higher compressive strength than the reference samples without coating?

25.  Provide specific values for the residual compressive strength of concrete samples with alkali-activated coatings after the fire resistance test?

26.  How much did the residual compressive strength decrease for the reference sample after three hours of the fire resistance test?

27.  What percentage of residual compressive strength was achieved for concrete samples with alkali-activated slag coatings after the fire resistance test?

28.  How did the thermal stability of concrete samples with alkali-activated slag coatings compare to those without coatings?

29.  What distinguishes the behavior of alkali-activated slag at high temperatures from ordinary Portland cement?

30.  Were there specific phases or changes in mineral composition that contributed to the increase in compressive strength?

31.  Why is it important to assess the performance of reinforced concrete structures with glass fiber polymer (FRP) bars in elevated temperatures?

32.  How did the bond strength between glass FRP bars and concrete change when exposed to elevated temperatures? What was the bond strength of the reference sample before exposure to elevated temperatures? How did the bond strength of glass FRP bars and concrete change when protected with alkali-activated slag coatings?

33.  The manuscript requires an explanation of the mechanisms underlying geopolymerization and hydration. To fill this gap, it is recommended to include a dedicated paragraph after the introduction section to clarify these processes. To support the authors, the inclusion of pertinent articles is suggested. Including an elucidation of the geopolymerization and hydration mechanisms is crucial to improving the manuscript's comprehensiveness. doi.org/10.3390/polym14153132; doi.org/10.1016/j.jmrt.2023.05.085; doi.org/10.1515/rams-2022-0330

Author Response

Hello,

Reviewer 5 Report

Comments and Suggestions for Authors

The present paper deals with the study of protective coatings for concrete. Slag and phosphogypsum were studied for producing coatings for fire-resistant applications. Samples of concrete were covered by coatings produced by using slag, phosphogypsum and sodium hydroxide solution as alkali activator. Sodium hydroxide was kept constant at 9.71%wt and the amount of phosphogypsum varied in each type of coatings (0%, 1%, 3%, 5%, 7% and 10%)

The paper fits the journal’s scope. References are appropriate, although they could be enriched with more.

The paper is extensive, with a full explanation of the results of the tests performed. Conclusions are consistent with the evidence and arguments presented and do they address the main question posed by authors in the introduction. Figures and tables are easy to interpret and understand.

The following comments are also suggested:

-In abstract, line 14, results for concrete samples reinforced with fiberglass polymer (FRP) rods where mentioned, without previously mentioned the preparation of them.

- paragraph 2 must be rewritten. For example, preparation of reinforced concrete samples is described in raw materials paragraph.

-Line 143: “The amount of phosphogypsum (0%, 1%, 3%, 5% and 10%) varied in each type of coatings”: there was also a coating with 7%wt phosphogypsum.

-Line 163: rewrite the sentence

- Why figure 7b does not show standard deviation.

- give some more details about the preparation of the coatings.

- how compressive strength of coatings was determined?

Author Response

Hello,

Round 2

Reviewer 1 Report

Comments and Suggestions for Authors

1. The equation is not numbered correctly, (3) appears first, and then there are formulas (1) and (2)

2. Reference 9 is different with others.

3. The newly added references have different style. 

4. The objective in the last paragraph of the Introduction should be enhanced. 

5. The first occurrence of the abbreviation should give the full name and explanation

Comments on the Quality of English Language

The quality of English language is quite acceptable. 

Author Response

Hello,

Reviewer 2 Report

Comments and Suggestions for Authors

The author has completed most of the revision work, but there are still the following problems that need to be further solved.

1. It is necessary to simplify the content of Introduction, and delete references that are weakly related to the research content of this paper.

When introducing phosphogypsum, I suggest citing relevant references and should not write the relevant parts of phosphogypsum at the end of Introduction.

The advantages of this study compared with other studies are still not reflected in the Introduction. It is suggested to introduce the relevant research on alkali-activated slag, at least citing previous your research results such as: https://doi.org/10.1016/j.conbuildmat.2018.06.213

2. It is suggested that the various processes in Fig.5 should be expressed in the form of pictures instead of words.

3. The sentence “The best results were obtained using an alkali activated 3S coating during all study periods” in Conclsuion should be revised, because “3S” is just a code name, not a number.

Author Response

Hello,

Thanks for your suggestions.

Reviewer 4 Report

Comments and Suggestions for Authors

Manuscript ID: materials-2669596

Title: ALKALI ACTIVATED SLAG COATINGS FOR FIRE PROTECTION OF OPC CONCRETE

Comments: The authors have diligently and effectively responded to all of the feedback and comments provided and have made the necessary revisions to their manuscript. As a result, the revised version of the manuscript meets the required standards for publication, and it can be accepted.

Author Response

Thank you very much.